# Personalized Assessment of Insomnia and Sleep Quality in Patients with Parkinson’s Disease

**DOI:** 10.3390/jpm12020322

**Published:** 2022-02-21

**Authors:** Ştefania Diaconu, Cristian Falup-Pecurariu

**Affiliations:** 1Department of Medical and Surgical Specialties, Faculty of Medicine, Transilvania University, 500036 Brașov, Romania; stefi_diaconu@yahoo.com; 2Department of Neurology, County Emergency Clinic Hospital, 500365 Brașov, Romania

**Keywords:** Parkinson’s disease, insomnia, sleep quality, assessment, personalized medicine

## Abstract

Sleep disturbances are more common in patients with Parkinson’s disease (PD) than in the general population and are considered one of the most troublesome symptoms by these patients. Insomnia represents one of the most common sleep disturbances in PD, and it correlates significantly with poor quality of life. There are several known causes of insomnia in the general population, but the complex manifestations that might be associated with PD may also induce insomnia and impact the quality of sleep. The treatment of insomnia and the strategies needed to improve sleep quality may therefore represent a challenge for the neurologist. A personalized approach to the PD patient with insomnia may help the clinician to identify the factors and comorbidities that should also be considered in order to establish a better individualized therapeutic plan. This review will focus on the main characteristics and correlations of insomnia, the most common risk factors, and the main subjective and objective methods indicated for the assessment of insomnia and sleep quality in order to offer a concise guide containing the main steps needed to approach the PD patient with chronic insomnia in a personalized manner.

## 1. Introduction

Several non-motor symptoms are known to affect the quality of life in patients with Parkinson’s disease (PD), both in the early and advanced stages. Sleep disorders are found in all stages of PD, including in the pre-motor ones. Neurodegenerative processes that affect the normal functioning of neurotransmitters and the various effects of antiparkinsonian drugs could be involved in the pathogenesis of sleep disorders in PD [1]. Insomnia is reported by almost half of PD patients, and it is significantly related to motor fluctuations and other non-motor features [2]. 

The diagnosis of insomnia is based on the definition criteria established by the International Classification of Sleep Disorders, 3rd edition: difficulties to initiate and/or to maintain sleep and/or early morning awakenings [3]. For chronic insomnia, the above symptoms should be experienced by the patient at least 3 times a week for a minimum of 3 months. The association of daytime symptoms as consequences of insomnia is usually mandatory to establish the diagnosis [3]. Insomnia is commonly reported by PD patients, regardless of the severity of the disease, with a reported prevalence of 37–83% [4,5]. Polysomnographic studies revealed that PD patients have a shorter total sleep time and lower sleep efficiency compared to controls [6]. A polysomnographic analysis of 50 PD patients showed prolonged sleep latencies in almost half of the patients (mean duration: approximate 22 min) and a mean total sleep time of approximately 5 h/night [7]. Regarding the duration of sleep during the daytime, a study that objectively measured napping using wrist actigraphy in 85 PD patients showed a mean nap time of 39.2 ± 35.2 min/day [8]. All subtypes of insomnia (derived from the main definition) can be identified in PD, with variations across PD stages [9]. According to the results of a study performed on 689 PD patients, sleep maintenance insomnia due to disrupted sleep was the most commonly encountered (81.54%), followed by early morning awakenings (40.4%) [10]. Sleep disturbances, especially insomnia and reduced sleep quality, affect the quality of life [11]. Moreover, worsening of sleep disturbances and other neuropsychiatric complaints may contribute to the progression of other non-motor symptoms [12]. Some of the motor and non-motor symptoms are interrelated; for instance, gastrointestinal dysfunction may lead to poor absorption of antiparkinsonian drugs and to worsening motor symptoms [13]. Certain sleep disturbances might influence disease-related disability as well [14]. A proper and careful assessment of insomnia and other comorbidities is therefore mandatory in order to choose the right therapeutic intervention for each patient. Depending on the main causes of insomnia, the therapeutic options may vary from recommending sleep hygiene or cognitive behavioral therapy to pharmacological options such as benzodiazepine and nonbenzodiazepine hypnotics [15].

## 2. Assessment

### 2.1. Clinical Interview

A thorough history taking is essential when evaluating a patient with complaints suggestive of insomnia. The anamnesis should be obtained from the patient but also from the bed partner or caregiver. It is important to highlight the most important subjective symptoms, the time of the night when these complaints occur (first part of the night or after the patient falls asleep) and information regarding sleep patterns and habits. Regarding this aspect, the physician should be interested in the consistency or not of a sleep schedule, naps during the day, the consumption of alcohol, caffeine or other energizing products, as well as the type of physical activity performed by the patient. The patient’s medication should be reviewed in search of insomnia as a side effect. Even if dopaminergic medication is known to induce daytime sleepiness, it can also be associated with insomnia [16]. According to a meta-analysis, levodopa, dopamine agonists, acetyl-cholinesterase inhibitors, and certain antidepressants may cause insomnia as an adverse effect [17]. Therefore, the medication regimen of the patient and the effects of polypharmacy should be carefully evaluated. Additionally, the medication and personal strategies used to alleviate insomnia should be assessed. 

The clinician should search for the associated non-motor symptoms that can impair sleep quality and might sustain insomnia (an easy method in this regard is to use the Non-Motor Symptoms Questionnaire [18], which is described below) and also for the motor features that might interfere with satisfactory nighttime sleep (tremor, rigidity, dystonia). It is also essential to assess the consequences of insomnia that are experienced by the patient (e.g., headache, fatigue, daytime sleepiness, depression, anxiety) and the effects of insomnia on daily life activities. As daytime sleepiness frequently occurs in PD patients and is associated with episodes of sudden onset of sleep that might potentially be dangerous [19], it is important to ask PD patients with chronic insomnia if they feel sleepy during the day. A quick evaluation method of this symptom is to use the Epworth Sleepiness Scale (ESS) [20], which is one of the most commonly used scales for the evaluation of daytime sleepiness in the general population and also in PD. The Movement Disorders Society (MDS) Task Force considers the ESS a “recommended” instrument to screen for daytime sleepiness in the PD population [21]. Cognitive decline might also be considered as a negative consequence of sleep disorders, including insomnia [22]. For further assessment of cognitive function, the Mini-Mental State Examination (MMSE) and the Montreal Cognitive Assessment (MoCA) [23] are among the tools most commonly used in the general population. The MoCA has been shown to be more sensitive than the MMSE in detecting cognitive impairment in patients with PD [24] and other neurodegenerative disorders, such as progressive supranuclear palsy or multiple system atrophy [25]. The MDS Task Force recommends the following scales for cognitive screening in PD: MoCA, Mattis Dementia Rating Scale 2nd Edition (DRS-2) [26] and Parkinson’s Disease-Cognitive Rating Scale (PD-CRS) [27]. 

A sleep diary is an easy method of assessing patients’ sleep patterns that is used in the general population [28] and in PD patients for clinical evaluation [29], for comparison with other evaluation methods or for monitoring therapeutic effects [30]. There are several designs of sleep logs, but generally, the following information should be recorded by the patient: the bedtime hour, the estimated time of sleep onset, total sleep duration, wake-up time and awakenings overnight, sleep quality and the presence of naps/physical exercises/medication/alcohol or caffeine intake during daytime [28]. The sleep diary could also be used in association with other objective measurements.

### 2.2. The Assessment of Specific Risk Factors for Insomnia Associated with PD

Risk factors that lead to insomnia or are known to aggravate insomnia in the general population (e.g., age, stress, mood, maladaptive lifestyle, behavioral and environmental factors [31]) may also be found in PD. According to recent studies, overall sleep disturbances in PD were associated with napping during the day, watching the clock repeatedly and staying in bed when not able to fall asleep [32], or other inadequate sleep habits [33]. In addition to these factors, there are some specific symptoms in PD patients that contribute to insomnia and should be carefully assessed by the clinician. 

#### 2.2.1. Motor Symptoms

Motor features that are persistent or worsen during the night are associated with frequent awakenings and therefore, with sleep-maintenance insomnia. Nocturnal hypokinesia and rigidity might impair mobility and turning in bed and, therefore, might lead to sleep disturbances such as insomnia and reduced sleep quality and efficiency [34,35]. The persistence of tremors or dyskinesia during nighttime may also contribute to sleep fragmentation and poor sleep quality [36]. Motor symptoms may interfere with sleep maintenance even in the early stages; nocturnal dystonia, cramps and tremor were the motor features most commonly associated with sleep dysfunction in drug-naïve PD patients [37]. These motor symptoms, increased muscular tension, sleep apnea, age and disease duration may contribute to the concept of sleep fragmentation [38]. According to recent polysomnographic research, sleep fragmentation has a high rate in PD patients and might be considered a promising marker of PD progression [39]. 

#### 2.2.2. Non-Motor Symptoms

Urinary dysfunction, especially nocturia, is a common symptom in the general elderly population and is a major factor that contributes to sleep disturbances [40]. Nocturia is a commonly encountered dysautonomic feature in PD patients, and it may correlate with subjective insomnia [41], frequent awakenings and insufficient total sleep time [2]. 

Another non-motor symptom that is significantly associated with sleep disturbances is pain. This symptom was found in almost half of PD patients, and the most reported type of pain was musculoskeletal pain [42]. Martinez-Martin et al. reported that any type of pain, but mostly the musculoskeletal subtype, was significantly correlated with overall sleep disorders [43]. Polysomnographic (PSG) studies demonstrated sleep fragmentation and modifications of sleep architecture in patients with PD and pain, characteristics which were less prominent in PD patients without pain [42]. An easy and robust method to assess pain is to use the King’s Parkinson’s Disease Pain Scale, which was demonstrated to be reliable and valid for the evaluation of this complex symptom in PD patients [44]. 

Among psychiatric non-motor symptoms, depression and anxiety were the most associated with sleep disturbances and reduced sleep quality. The severity of insomnia is correlated with the severity of depression in PD patients [45]. Moreover, an interconnection between depression, anxiety and pain may be found in conjunction with sleep disturbances and poor sleep quality in PD patients [46,47].

#### 2.2.3. Other Associated Sleep Disorders

Restless legs syndrome (RLS), defined as discomfort in lower limbs that induces the need for movement and occurs during the night and during periods of immobility, is found in higher rates among PD patients compared to the general population [48]. RLS in association with periodic limb movements may induce arousal and disrupt normal sleep continuity, contributing to chronic insomnia [48,49]. The clinical diagnosis of RLS is based on the International Restless Legs Syndrome Study Group (IRLSSG) criteria [50]. There are various screening scales that can help the clinician to better assess these symptoms [51]. 

Sleep-disordered breathing (SDB) and mainly obstructive sleep apnea (OSA) may contribute to a reduced quality of sleep. Difficulties maintaining sleep are more common in patients with OSA than in healthy controls [52], and there is evidence that more than half of OSA patients without treatment have chronic insomnia [53]. In PD patients, the prevalence of OSA is approximately 62% [54]. Sobreira-Neto et al. found that PD patients with OSA may present lower rates of chronic insomnia (probably due to their reduced insight of sleep onset latency caused by sleep deprivation), but they also show reduced time spent in the N3 sleep stage and higher numbers of arousals compared to PD patients without OSA [55]. 

REM sleep behavior disorder (RBD), a parasomnia with “dream enacting”, expressed by abnormal movements instead of muscular atonia during the REM sleep stage, is commonly associated with PD and with other neurodegenerative disorders. Longitudinal studies, including patients with idiopathic RBD, show significant correlations with neurodegenerative disorders and even high rates of phenoconversion [56]. RBD in PD patients causes reduced quality of sleep and also induces other consequences, such as cognitive impairment and autonomic dysfunctions [57,58]. RBD is associated with disrupted sleep architecture, explained by lower percentages of N2 and N3 sleep and a high periodic limb movement index [59]. 

### 2.3. Rating Scales and Objective Assessment of Insomnia and Quality of Sleep in PD Patients

#### 2.3.1. Multidomain Scales or Questionnaires Designed to Evaluate Non-Motor Symptoms in PD, including Insomnia

Many of these scales contain items for screening and/or assessment of the severity of sleep disturbances. 

*The Movement Disorders Society Unified Parkinson Disease Rating Scale (MDS-UPDRS)* is one of the most commonly used scales for the complex evaluation of various features of PD. It contains four parts, and part I is designed for the assessment of non-motor complaints. There is one question regarding insomnia and one question about daytime sleepiness. For all the items in this first part of the scale, the answers can be chosen between “0 = normal” and “4 = severe” [60].

*The Non-Motor Symptoms Questionnaire (NMSQ)* [18] has proven its efficiency in screening non-motor features of patients with PD. It contains 30 questions with simple answer choices (“yes”/“not”), and five of them are dedicated to the assessment of sleep: the difficulty of staying awake in certain circumstances, the difficulty of falling asleep and maintaining sleep, vivid dreams or nightmares, speaking or having abnormal movements during sleep and unpleasant sensations in the lower limbs during nighttime or rest associated with the need to move. This questionnaire is very easy for the patient to complete. It offers an accurate overview of the main symptoms, but it does not offer any information regarding the severity or the frequency of these symptoms. There are no items addressing SDB.

*The Non-Motor Symptoms Scale (NMSS)* [61] was created to complete and deepen the information obtained with the NMSQ. It is completed by the examiner. It takes a longer time for scoring than the NMSQ, but it provides data regarding symptom frequency and severity. For all the 30 items, the severity of the symptoms is evaluated from “0 = none” to “3 = severe”, and the frequency is scored from “1 = rarely” to “4 = very frequent”. The total score for each item is obtained by multiplying its severity by its frequency. The sleep/fatigue domain is comprised of items about insomnia, daytime sleepiness, RLS and fatigue.

*The International Parkinson and Movement Disorder Society Non-Motor Rating Scale (MDS-NMSS)* [62] is a reviewed version of the NMSS. It is administered by the examiner; it contains 52 items about non-motor symptoms, and the scoring process is similar to the NMSS: the frequency of the symptom is rated from “0 = never” to “4 = majority of time” and the severity is chosen between “0 = never” to “4 = severe”. The product (frequency × severity) is calculated for each question. The “sleep and wakefulness” domain contains six questions about insomnia, RBD, excessive daytime sleepiness (EDS), restlessness, periodic limb movements and SDB. 

#### 2.3.2. Specific Scales or Questionnaires Designed to Evaluate Insomnia and Sleep Quality

*The Parkinson’s Disease Sleep Scale (PDSS)* [63] is a self-assessment rating scale encompassing 15 questions which assess general aspects of sleep/daytime symptoms within one (previous) week: items related to insomnia, restlessness, hallucinations, bladder dysfunctions, tremor, dystonia, overall quality of sleep, and EDS. It was designed to assess sleep disturbances in the PD population, with each item being evaluated based on a visual analogue scale (VAS) ranging from 0 (never/excellent) to 10 (always/awful). A cutoff of 82/83 is considered suggestive for sleep disturbances, and the maximum score is 150 points, representing the worse clinical picture [64]. 

Regarding the psychometric properties of the PDSS, in the original study, the PDSS was applied on 280 adults (143 PD patients in different stages of severity and 137 healthy controls). Test-retest reliability was very high; good internal consistency (Cronbach’s alpha, 0.77) and high repeatability were also observed [63]. Floor and ceiling responses were low (1%) [58,60]. High scores on the PDSS regarding EDS correlated with low scores on the ESS, which is the main scale used to assess EDS [63]. The PDSS showed a good correlation with SCOPA-S scores as well [64]. 

Regarding the strengths of the PDSS, it was developed as a brief, easy-to-use, reliable bedside instrument to screen for sleep symptoms in PD patients. Based on the results, the clinician may have insights regarding the severity of the sleep complaints. The items addressing insomnia might help discriminate between the types of insomnia or the possible causes (e.g., nocturia, tremor). The MDS Task Force classified the PDSS as a “recommended” tool to assess the existence and the severity of sleep disorders in PD [21]. It was shown that the PDSS could discriminate between PD and controls and also within PD severity levels and duration [64]. 

The PDSS is in the public domain and has been validated in several languages (Spanish, Japanese, Portuguese), with good clinimetric properties [65,66,67]. It was also widely used in several clinical trials in the PD population—e.g., evaluating the effectiveness of rotigotine [68]). 

Regarding the weaknesses of the PDSS, even if the VAS is considered a simple method of assessing the level of severity of symptoms, it may be necessary to first inform the patient or caregiver how to apply this scoring system correctly.

The scale has only one question regarding EDS; therefore, it does not represent a proper tool to assess daytime symptoms. There are no questions related to other sleep disturbances, like sleep apnea or RBD; regarding RLS, there is only one question about “restlessness” of the arms or legs that might be related to RLS symptoms, but the mandatory criteria for RLS diagnosis are not fulfilled. The proposed timeframe is the previous week. 

Based on the experience gathered from the administration of the PDSS, a revised version, the *Parkinson’s Disease Sleep Scale (PDSS-2),* was designed with the intention of exploring several aspects of sleep in the PD population that were not evaluated in the first version. It was also intended to be a useful tool to assess the effects of treatment on sleep disturbances [69]. 

The PDSS-2 includes 15 questions for the self-evaluation of sleep symptoms, addressing sleep quality, insomnia, restlessness, nightmares/hallucinations, bladder problems, motor features like rigidity or tremor, pain and breathing difficulties. The VAS scoring system was replaced with a grading system of symptom severity from 0 (never) to 4 (very frequent), with a maximum score of 60, indicating severe nocturnal sleep disturbances [69]. A score of 15 or above was considered the cutoff for poor sleepers [69,70]. 

Regarding the psychometric properties of the PDSS-2, the total score was evaluated, as were the scores for the three subscales (motor problems at night; PD symptoms at night; sleep specific disturbances) in order to establish the clinimetric characteristics. The PDSS-2 was validated in order to investigate nighttime impairments for the PD population, with findings demonstrating satisfactory reliability (Cronbach’s alpha coefficient of 0.73 for the total score and few variations for the sub-scores), good internal consistency for most of the items (>0.30), and high test-retest reliability within 1–3 days (ICC of 0.80 for the total score) [69]. The test-retest reliability for a longer timeframe (1 month) was evaluated in another study, and it was considered acceptable (ICC of 0.799 for the total score) [71].

Regarding the strengths of the PDSS-2, it was shown that the PDSS-2 is a brief, easy-to-use and easy-to-administer self-rating scale useful for both screening the existence of sleep symptoms and grading their severity. Compared to the previous version, it is easier for patients with PD to understand and complete the PDSS-2 scale due to its Likert scoring system. It has good discriminative power between the grades of disease severity as evaluated with the Hoehn and Yahr scale. PDSS-2 is an assessment tool belonging to the public domain, which was validated and translated into several languages (German [69], Spanish [72], Italian [73], and Chinese [74]). It has been widely used in prevalence studies to assess sleep symptoms and their associations with several other symptoms or objective investigations (e.g., the presence of sleep disturbances and their correlations with brain MRI morphometry [75]), and it was used in clinical trials to evaluate the efficiency of medication on sleep [76].

Regarding the weaknesses of the PDSS-2, it focuses on the existence and severity of nighttime symptoms and therefore is not a proper tool to investigate their diurnal consequences, such as EDS. Unlike the previous version, the PDSS-2 has more clear questions related to RLS, and it also assesses the existence of breathing disturbances, but these items are not precise enough to diagnose RLS or OSA. A caregiver might improve the accuracy of the answers for some of the items, like for those related to awakenings during the night or difficulties turning in bed (as the patient might underestimate the existence/severity of these issues). The proposed timeframe is the previous week. 

*The Scales for Outcomes in Parkinson’s Disease—Sleep (SCOPA—Sleep)* is a 12-item self-rating scale designed to specifically evaluate nighttime sleep and daytime consequences in PD patients [77]. The questionnaire is structured in three parts. The first part consists of five items representing nighttime-specific (NS) disturbances that the patient might have experienced in the previous month (most of them concerning different types of insomnia). To answer these 5 questions, the patient chooses the answer which fits best from 0 (not at all) to 3 (a lot). The maximum score for this part is 15, indicating severe nighttime impairments (cutoff: 6/7) [64,77]. The second part is composed of only one question regarding the quality of sleep during the night; there are seven response options, ranging from “very well” to “very badly”. There is no numeric scoring for this part. The last part contains six items to evaluate the daytime symptoms (DS) in the previous month, including EDS and the existence of sudden onset of sleep. The scoring for each item varies from 0 “not at all” to 3 “very much”, with a maximum of 18 and a cutoff of 4/5 indicating daytime disturbances [77]. 

Regarding the psychometric properties of SCOPA-sleep, it shows high internal consistency for both the NS and DS subscales (Cronbach’s alpha: 0.88 and 0.91, respectively) and good test-retest reliability (ICC: 0.94 for the NS subscale and 0.89 for the DS subscale). There were robust correlations between the DS subscore and ESS; the subscores of the NS part correlated with the PDSS and PSQI [77]. The floor and ceiling effects are absent [64]. 

Regarding the strengths, this scale is a brief, easy-to-administer rating tool with good internal consistency and reproducibility which can be used for screening and quantifying nighttime and daytime symptoms (like sudden onset of sleep) in PD patients. The MDS task force indicated SCOPA-sleep as a “recommended” scale for the aforementioned purposes [16]. The scale has been translated into several languages, taking part of the public domain. Like other scales designed to assess sleep in PD, SCOPA-sleep was useful to analyze the effect of various therapeutic options on sleep [78], and it was also used for monitoring symptoms in longitudinal studies [2].

Regarding weaknesses, even if the scale is designed to screen for possible nighttime symptoms, SCOPA-sleep lacks questions addressing nocturia, RBD, RLS or OSA. There are no questions addressed to the caregiver. 

*The Pittsburgh Sleep Quality Index (PSQI)* [79] represents a self-rating tool designed to assess sleep in the general population, with the timeframe being the previous month. The first four items are dedicated to sleep habits (like usual bedtime, the perceived sleep latency, and number of hours of sleep per night). This is followed by questions related to possible causes of sleep disturbances (e.g., insomnia, breathing difficulties, pain) and questions about sleep quality, use of sleep medication, difficulties staying awake during daytime activities, and difficulties maintaining enthusiasm in daily activities. The PSQI has an additional five informative questions for the bed partner, which do not accumulate to the final score. For each item, the answers can be scored from 0 to 3 (no impairment/severe impairment). Based on the type of sleep problem addressed, the results can be grouped into seven compounds. The total score reaches a maximum of 21 points (indicating severe sleep disturbances), and a score of more than 5 points (for the total items) was considered an indicator for “bad” sleepers [79]. For PD patients, a more appropriate cutoff was considered 8/9 [77]. 

Regarding the psychometric properties of the PSQI, in the original study published in 1989, it was demonstrated to have high internal consistency and homogeneity (Cronbach’s alpha: 0.83) [79]. Test-retest reliability was high for a short interval (2 days apart), and it remained high for a longer timeframe considering the majority of the subscores and the total score (overall test-retest correlation coefficient: 0.87) [80]. The PSQI showed correlations with PSG only regarding sleep latency, but it has strong correlations with the SCOPA-Sleep scale [81]. 

Regarding its strengths, the PSQI is in the public domain, and it is used to assess sleep in the general population and in PD patients [54,82]. Even if not specifically validated for PD, the PSQI is considered by the MDS Task Force to be a “recommended” tool to investigate sleep in the PD population [21]. Furthermore, it is a commonly used scale to evaluate the occurrence of sleep disturbances in primary insomnia, dementia and other movement disorders [81]. It has been largely translated into several languages, and it is also useful for monitoring the impact of various interventional strategies on sleep parameters [83,84]. 

Regarding its weaknesses, even if it covers a large spectrum of sleep disturbances that might occur, the PSQI has limited power to assess some conditions, such as OSA or RBD, and has no items designed to evaluate RLS. The questions addressed to the bed partner can help the investigator to complete the picture of the patient’s overall sleep disturbances, but the data is not included in the total score; consequently, the global severity may be underestimated. The scoring system is complex, and some additional time should be considered for this aspect; the investigator has a guide with instructions for scoring.

A summary of the main scales used to assess insomnia and sleep quality in PD patients is presented in Table 1.

### 2.4. Objective Methods to Assess Insomnia

*Actigraphy* is a non-invasive method that is able to investigate several sleep parameters based on recording limb activity via accelerometers [85]. The device is worn on the non-dominant hand for a minimum of one week, and the results are interpreted together with a sleep log. Actigraphy is a useful method for the assessment of insomnia in the general population, as patients have the tendency to overestimate their sleep onset latency and to have a lower perception regarding the total sleep time [86]. On the other hand, some studies have demonstrated that PD patients might actually have a more accurate perception of their sleep problems [87]. Actigraphy was validated in patients with insomnia in the general population [88,89,90,91] and also demonstrated accurate results in evaluating sleep quality in PD patients compared to other subjective measures [29]. Actigraphy has several limits, though, as it cannot offer information about sleep stages, and it may overestimate the total sleep time if the patient remains still in bed without moving, as the recorder misinterprets immobility as sleep [91]. Therefore, actigraphy is best indicated for characterizing sleep disruptions and not to certify sleep initiation insomnia [85].

*The Parkinson’s KinetiGraph (PKG)* is a device using wearable sensors (accelerometers) to record movements in order to offer data regarding several motor parameters in PD. It can also provide relative information regarding sleep parameters, as a period of immobility detected for at least 14 min is considered as an episode of sleep. The presence of interruptions of immobility during the night might be interpreted as awakenings or abnormal movements caused by sleep disturbances (RLS, RBD, etc.) [92]. Klingelhoefer et al. reported that the immobility and mobility states recorded by the PKG might correspond to sleep/awakening periods during nighttime, and the recorded sleep parameters correlate with other subjective measures of sleep [92]. The information obtained with the PKG changed the therapeutic decision in almost one-third of PD patients and improved communication with the neurologist in the majority of cases [93]. Comparative studies with polysomnography indicated that the periods of immobility that were identified with PKG during daytime correspond in approximately 85% with sleep periods confirmed with PSG [94]. Considering this, PKG might be a useful tool to investigate sleep onset and maintenance insomnia [92] and to integrate the information with the concomitant objective measures of the motor symptoms (bradykinesia, tremor, dyskinesia and fluctuations) [95,96]. PKG might also provide information regarding sleep quantity and quality, with significant correlations with subjective measures, but it cannot establish the sleep stages only based on the immobility data recorded [92], nor can it establish other events such as OSA or periodic limb movements [97]. It is a reliable tool that should be used together with a thorough history and clinical examination [93].

*Polysomnography (PSG)* is considered the ‘gold standard’ assessment tool for sleep disorders, as it can evaluate in an objective manner the sleep stages, sleep architecture and the normal and abnormal events during sleep. Several studies have been conducted in order to evaluate the sleep differences between PD patients and healthy controls. In PD patients, most of the PSG studies revealed more awakenings in PD patients, but no differences in sleep stages 1, 2 and the slow-wave sleep stage were observed in comparison to controls [98]. Most of the PSG studies did not demonstrate an increased rate of periodic limb movements during sleep in PD patients, nor a certain association with obstructive sleep apnea [98]. The contribution of nocturia to disrupted sleep and poor sleep quality was also demonstrated objectively by PSG evaluation [99]. Regarding the suspicion of RBD in PD patients, the clinical interview of the patient and PD partner might underestimate the occurrence of the abnormal motor behavior during sleep; therefore, PSG is necessary for the correct diagnosis of RBD [100]. However, PSG does not take part in the routine assessment of insomnia, as it has several limits—it is laborious and it requires trained clinicians and special conditions for the assessment (sleep lab). It is neither useful nor recommended to diagnose insomnia, but it can be necessary to rule out other conditions that might induce and perpetuate insomnia, such as SDB, RBD, and periodic limb movements [101]. 

## 3. Personalized Medicine and the Assessment of the PD Patient with Insomnia and Impaired Quality of Sleep

The concept of personalized (precision) medicine emphasizes the need for multidimensional approaches to the PD patient considering the complexity of this disorder. Several factors should be reviewed in order to establish tailored management strategies: genomics, pharmacogenetics, personality, lifestyle, comorbidities, etc. [102]. For instance, the comorbidity of respiratory disorders/sleep apnea, which is related to excessive daytime sleepiness, sleep fragmentation, anxiety and memory difficulties, represents one of the interrelated clinical situations addressed by personalized medicine [103]. In that case, the proper assessment of sleep-maintenance insomnia could reveal an underlying respiratory problem and consultation with a pulmonologist would be necessary. Taking into account the various factors known to be associated with insomnia and the particularities of the sleep disturbances in PD, an individualized approach is therefore mandatory in order to better characterize sleep in PD and to develop adequate management strategies. We propose an algorithm for the personalized assessment of PD patients with insomnia and impaired sleep quality, which is shown in Figure 1. In this regard, the clinician should always approach the patient with sleep disturbances by asking for more details about the main complaints (the information obtained from a caregiver could be valuable). A full general and neurological exam should be performed; a proper examination of the motor symptoms should include assessment using the UPDRS part III & IV. As the non-motor symptoms have strong connections with insomnia and poor sleep quality, the NMSQ can be used as a screening tool that is brief and easy to apply. To better understand the patient’s pre-sleep habits and symptoms, several aspects should be asked (for instance, sleep patterns during day and night, the intake of coffee or alcohol, if the patient leads a sedentary lifestyle, etc.). A sleep diary may bring valuable information in this regard, and the patient should be informed regarding how to overcome his or her bad sleep habits. The side effects of medications should be reviewed and changed accordingly. Once the clinician identifies certain symptoms that occur before sleep onset (for instance, RLS), the next step is to evaluate the severity of these symptoms and start a treatment (in this case, IRLS might be a useful tool for severity grading and monitoring). If the patient complaints about poor sleep quality, frequent awakenings during nighttime and difficulties falling back asleep, the neurologist should try to identify and treat the cause(s), considering the common association with motor symptoms (nocturnal cramps, tremor, dyskinesia, etc.), non-motor symptoms (e.g., pain, nocturia) and other comorbidities (RLS/ PLMS, SDB, RBD). In the context of a busy medical practice, we strongly recommend the use of standardized scales for non-motor evaluation (NMSQ) and for overall sleep assessment (considering their main indications, advantages, and disadvantages—see Table 1). Daytime consequences of insomnia and poor quality of sleep should be asked about in order to appreciate the magnitude of the sleep complaints. For instance, if the patient or the clinician suspects that poor quality of sleep may be associated with memory and attention problems, a cognitive screening test such as the MMSE or MoCA can offer supplementary information. In some cases, when the causes or the consequences of insomnia and poor sleep quality are difficult to identify or are resistant to the recommended treatment, further objective assessment should be indicated. Wearable devices might be more convenient for the patient. They are useful for obtaining objective measurements of sleep parameters and motor function and can record information for a longer time (at least 1 week). PSG, on the other hand, should be indicated only in particular circumstances, for instance, when the diagnosis is uncertain or when other associated conditions such as SBD or RBD are suspected. Considering the many interconnected aspects of sleep disorders in PD patients, we suggest that a comprehensive assessment of sleep parameters and associated factors may be the key to personalized and successful management of these disturbances.

## 4. Conclusions

There are several aspects of sleep that should be carefully examined when investigating the PD patient with insomnia. Many behavioral factors, as well as the associated motor and non-motor symptoms, are interconnected with sleep disturbances and poor quality of life. An easy and methodical approach is to start from the main complaint and then assess the habits and symptoms before sleep, then the symptoms during sleep and the consequences during the day. There are several useful scales and questionnaires designed to help the clinician identify the main complaints and to grade their severity. When in doubt, further objective assessment methods should be recommended, such as actigraphy, the Parkinson KinetiGraph or polysomnography. A personalized approach to the PD patient with sleep disturbances would be therefore much effective in establishing the proper therapeutic strategies that can help improve the quality of life of these patients.

## Figures and Tables

**Figure 1 jpm-12-00322-f001:**
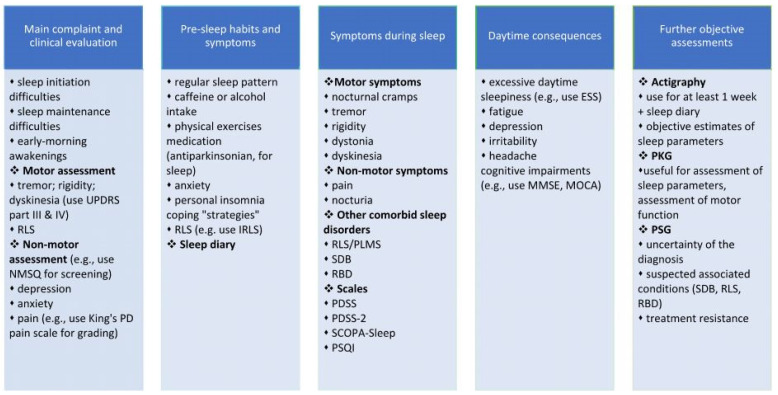
Proposed clinical interview components and subjective and objective methods for the personalized assessment of insomnia and sleep quality in PD patients. ESS, Epworth Sleepiness Scale; IRLS, International Restless Legs Syndrome Study Group rating scale; MMSE, Mini-Mental State Examination; MOCA, Montreal Cognitive Assessment; NMSQ, Non-Motor Symptoms Questionnaire; PD, Parkinson’s disease; PDSS, Parkinson Disease Sleep Scale; PKG, Parkinson’s KinetiGraph; PLMS, periodic limb movements of sleep; PSG, polysomnography; PSQI, Pittsburg Sleep Quality Index; RBD, REM sleep behavior disorder; RLS, restless legs syndrome; SCOPA, Scales for outcomes in PD; SDB, sleep-disordered breathing; UPDRS, Unified Parkinson Disease Rating Scale.

**Table 1 jpm-12-00322-t001:** Main characteristics of the most commonly used scales for the assessment of insomnia and sleep quality.

Scale Name	Designed For	Nr. Items	Approximate Completion Time	Short Description	Time Frame	Cutoff	Advantages	Disadvantages
PDSS	General sleep assessment in PD patients EDS	15	10 min	Self-assessment Items regarding insomnia, restlessness, various symptoms during nighttime, EDS Evaluation based on VAS (0–10) Maximum: 150 points = no sleep problems	Previous week	82/83	Brief, easy to administer bet used to screen/assess the severity of nocturnal/diurnal sleep disturbances	VAS scoring requires instruction to complete No items addressing RBD, RLS, or breathing disturbances
PDSS-2	General sleep assessment in PD patients PD symptoms during sleep	15	10 min	Self-assessment Items regarding insomnia, restlessness, various symptoms during nighttime, including motor features and breathing difficulties Evaluation based on Likert scale (0–4) Maximum: 60 points = severe sleep disturbances	Previous week	≥15	Brief, reliable, precise Used to screen/assess the severity of nighttime complains	No daytime symptoms assessment The insight of a caregiver may be necessary for some items
SCOPA—sleep	Nighttime symptoms, quality of life and daytime symptoms in PD patients	12	5–10 min	Self-assessment One part for assessing nighttime symptoms (mainly insomnia) One question regarding sleep quality One part for assessing daytime symptoms (EDS, sudden onset of sleep)	Previous month	6/7 for night symptoms 4/5 for day symptoms	Brief, easy to administer Useful for screening and grading nighttime + daytime symptoms	No questions for SDB, RLS, RBD nocturia
PSQI	General sleep assessment EDS	19	5–10 min for completing5 min for scoring	Self-assessment Sleep habits evaluation; insomnia, various causes for sleep disturbances; EDS Scoring from 0 (no difficulties) to 3 (severe) based on a guide for investigator Maximum: 21 points = severe sleep disturbances	Previous month	>5 in the general population; >8 in PD patients	Good insight into quality of sleep, sleep habits and causes of sleep disturbances	Questions related to SDB, RBD—ambiguous No questions for RLS Information from bed partner not accounted for in the total score Additional time for scoring (which is complex)

EDS, excessive daytime sleepiness; PD, Parkinson’s disease; PDSS, Parkinson Disease Sleep Scale; PSQI, Pittsburg Sleep Quality Index; RBD, REM sleep behavior disorder; RLS, restless legs syndrome; SCOPA, Scales for outcomes in PD; SDB, sleep-disordered breathing, VAS, Visual Analogue Scale.

## Data Availability

Not applicable.

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
