# Peer review of "Personalized Assessment of Insomnia and Sleep Quality in Patients with Parkinson’s Disease"

_jpm, 2022, doi:10.3390/jpm12020322_

Round 1

Reviewer 1 Report

Diaconu and Falup-Pecurariu present a review of current knowledge in the field of sleep and Parkinson’s disease (PD). It is focused on the role of insomnia and sleep quality, their assessment methods, and explores personalized approach to their evaluation in this population. Overall, the article serves its purpose and represents an updated source of information on the topic. However, there is some need for improvement in the following points:

  1. The authors will need to improve the English language, especially in the first half of the article, as across the text there is an unusual word usage, confusing terms. Below are several examples. In the Introduction section the word “fluctuations” in “All subtypes of insomnia (derived from the main definition) can be identified in PD, with fluctuations across PD stages [6].” sentence can be misinterpreted as PD motor fluctuations, better to substitute by “variations” or similar. Also, the “sleep impairments” term is not a common term, and is more confusing, so I would suggest substituting it across the whole text with “sleep disturbances”. Another: “… is significantly related to motor fluctuations and with other non-motor features…” remove “with”. In the Introduction section the sentence: Worsening of sleep impairments (disturbances) and of other (remove “of other”) neuropsychiatric complaints may contribute to the progression of other non-motor symptoms.” Also, “The moment of the night” – maybe the part of the night or the time of the night? Global (??) sleep disorders – “overall” or “similar”.
  2. In the article there is a clearcut accent on the link between insomnia and excessive daytime sleepiness, which seems to be overestimated as a daytime consequence of insomnia. The authors should be more comprehensive in this regard and review also other connected factors (OSA, pharmacotherapy, etc.). In some parts, it seems that the term insomnia is used instead of poor sleep quality, which is not the samse and needs separate discussion.
  3. The subheading 3 on Personalized approaches in the assessment of the PD patient with insomnia and poor quality of sleep is good but still short. Despite the inclusion of the Figure 1, which summarizes the reviewed information, still it needs some expansion, considering the main concept of the journal.
  4. The References section contains no numbering, it was really hard to follow the references manually putting the numbering. Please, correct this carefully.

Reviewer 2 Report

This paper is a concise, well-written review tackling sleep quality (more precisely Insomnia) in Parkinson’s disease (PD). Sleep alterations are indeed deleterious and frequent in PD patients and their management improves the quality of life in patients.

The authors discussed nicely the clinical recommendations to assess sleep alterations in PD and then enumerated the risk factors associated with insomnia in PD patients. The impact of medication was not discussed in this section and should be added to a complete list.

The authors then offered a detailed critical evaluation of the questionnaire used to assess sleep issues in PD. On page 4 (PDSS section); the max score of 150 represents the worse clinical picture not the absence of sleep symptoms. Please correct.

Insomnia as defined and assessed in PD refers to nighttime difficulties to initial and/or maintain sleep. What about the total 24h sleep amount? If we take into account daytime naps, do PD patients suffer from a total decrease or increase of total sleep time? Please cite relevant literature.

In relation to this last point, on page 9, PSG section is really poor and did not include the studies (though still few) that recorded PSG in PD patients.

References in the list are not enumerated as in the text.

Round 2

Reviewer 2 Report

Nothing.